# Alpha-Synuclein as a Prominent Actor in the Inflammatory Synaptopathy of Parkinson’s Disease

**DOI:** 10.3390/ijms22126517

**Published:** 2021-06-17

**Authors:** Antonella Cardinale, Valeria Calabrese, Antonio de Iure, Barbara Picconi

**Affiliations:** 1Laboratory Experimental Neurophysiology, IRCCS San Raffaele Pisana, 00166 Rome, Italy; antonella.cardinale@sanraffaele.it (A.C.); valeria.calabrese@sanraffaele.it (V.C.); antonio.deiure@sanraffaele.it (A.d.I.); 2Department of Neuroscience, Università Cattolica del Sacro Cuore, 00168 Rome, Italy; 3Department of Medicine, Università degli Studi di Perugia, 06123 Perugia, Italy; 4Department of di Scienze Umane e Promozione della Qualità della Vita, Università Telematica San Raffaele, 00166 Rome, Italy

**Keywords:** synaptopathy, α-synuclein, dopamine, neuroinflammation, immune system

## Abstract

Parkinson’s disease (PD) is considered the most common disorder of synucleinopathy, which is characterised by intracellular inclusions of aggregated and misfolded α-synuclein (α-syn) protein in various brain regions, and the loss of dopaminergic neurons. During the early prodromal phase of PD, synaptic alterations happen before cell death, which is linked to the synaptic accumulation of toxic α-syn specifically in the presynaptic terminals, affecting neurotransmitter release. The oligomers and protofibrils of α-syn are the most toxic species, and their overexpression impairs the distribution and activation of synaptic proteins, such as the SNARE complex, preventing neurotransmitter exocytosis and neuronal synaptic communication. In the last few years, the role of the immune system in PD has been increasingly considered. Microglial and astrocyte activation, the gene expression of proinflammatory factors, and the infiltration of immune cells from the periphery to the central nervous system (CNS) represent the main features of the inflammatory response. One of the actors of these processes is α-syn accumulation. In light of this, here, we provide a systematic review of PD-related α-syn and inflammation inter-players.

## 1. Introduction

The reduction of striatal dopaminergic neurons triggers motor symptoms that include bradykinesia, uncontrollable tremor at rest, postural impairment, and rigidity, which together characterise Parkinson’s disease (PD) as a movement disorder [1,2]. Neurodegeneration in the *Substantia Nigra pars compacta* (SNpc) leads to a marked decrease of dopamine (DA) levels in the synaptic terminals of the dorsal striatum and the consequent loss of nigrostriatal pathway, which allows PD to be described as a synaptopathy [3,4]. Synaptopathy is linked to α-synuclein (α-syn), a small, soluble protein encoded by the *SNCA* gene on human chromosome 4 [5,6], which is physiologically mainly localised in the presynaptic nerve terminals [7], the mitochondrial-associated membrane (MAM) [8], in which its overexpression increases the extent of contact sites and downregulates MAM activity [9,10,11,12], and in the nucleus [13,14,15,16].

α-syn accumulation compromises the fusion and clustering activity of the synaptic vesicles [17] and then influences neurotransmitter release, inducing the death of nigrostriatal neurons [18]. The transmission of α-syn pathology crosses different brain regions [19], though the impacts of extracellular α-syn on synaptic activity remains largely unknown [20,21]. Pacheco and colleagues [22] showed that extracellular α-syn oligomers facilitate the perforation of the neuronal plasma membrane, increasing its conductance and the influx of both calcium (Ca^2+^) and glucose, explaining in part the synaptotoxicity observed in PD. α-syn occurs in a dynamic balance between the monomeric and oligomeric forms, which are not easily prone to form fibrils under physiological conditions. Identifying the most toxic species, between fibrils and oligomers, has been difficult. There is evidence that the formation of fibrils mediates α-syn toxicity [23]. On the contrary, oligomeric forms are considered the most toxic species at the synapses [24], where they impair long term potentiation (LTP) and increase basal synaptic transmission through a mechanism dependent on N-Methyl-D-Aspartate (NMDA) receptor activation [25,26,27,28].

PD has a multifactorial aetiology. Indeed, the possible causes depend both on genetic and environmental factors that engage several biological mechanisms and processes, such as the cited α-syn misfolding, mitochondrial dysfunction, oxidative stress, synaptic plasticity, and neuroinflammation. Neuroinflammation, in recent years, has assumed a central role in the pathophysiology of PD and other neurodegenerative diseases. Evidence from post-mortem brains of PD patients, as well as in in vitro and in in vivo models, has highlighted the inflammatory contribution to the disease’s neuropathology [29,30,31,32,33]. In 1988, McGeer and collaborators disclosed the presence of HLA-DR^+^ microglia (macrophages) in the SN of idiopathic PD patients through immunohistochemical staining [34]. This finding proved the existence of the reactive microglia state around dead and dying dopaminergic neurons. Moreover, these macrophages exhibited phagocytic activity, as demonstrated by the presence of melanin detritus inside them [34]. This study, like others in the recent literature, suggested the involvement of immune system alteration in PD. Moreover, GWAS (genome-wide associations studies) analyses have unveiled the contribution of both innate and adaptive immune responses [35,36,37,38]. Furthermore, data support that oligomeric and fibrillary α-syn forms can activate microglial cells [39,40], suggesting a clear role for this protein in the central inflammation in PD affecting neuronal homeostasis through the modulation of microglia function, which could be either protective or detrimental in PD.

In this review, we will address the critical role of α-syn in relationship with the microglia and astrocytes, as immune resident brain cells, in the context of PD synaptopathy.

## 2. Synaptopathy

Synapses work within intricate neural networks to coordinate the neuronal activity involved in learning, memory, and behaviour. Therefore, damage to the synapses will have detrimental consequences [41,42]. Synapses have a presynaptic compartment, which includes the axon terminal and their protein machinery involved in neurotransmitter release [43]. After the exocytosis of presynaptic vesicles, neurotransmitters spill out and diffuse in the extracellular synaptic cleft to reach a specific postsynaptic compartment, which is composed of protein machinery that receives and transmits the signals induced by the neurotransmitters [44].

Therefore, early or late synaptic dysfunction drives a group of neurological disorders named synaptopathies [41,45]. This category of diseases includes neurodegenerative disorders, such as PD, which is characterised by a progressive degeneration of dopaminergic neurons located in the midbrain and a decline in cognitive and behavioural functions [4]. This pathological condition has been attributed to the accumulation of α-syn aggregates in neurons having detrimental consequences for neuronal integrity [23,46].

In the first part of this review, we focus on emphasising the current knowledge on synaptopathy and synaptic dysfunction induced by α-syn, and its role in the early stages of PD progression. Then, we describe the relevant role of the glial inflammatory reaction in the pathophysiology of PD, related to the detrimental effects of α-syn.

### 2.1. α-Synucleinopathy

The neuropathological hallmarks of PD are the progressive loss of dopaminergic neurons, specifically in the SNpc, and the presence of intraneuronal α-syn inclusions, termed Lewy bodies (LBs) and Lewy neurites (LNs) [47,48,49]. Synaptopathy is closely linked to α-syn, a small soluble protein that it is physiologically localised in the presynaptic terminals, compromising the fusion and clustering activity of the synaptic vesicles, and then influencing neurotransmitter release [50,51,52]. The primary presence of α-synucleinopathy is in the synaptic terminals, which results in an early synaptic impairment that precedes axon degeneration, with a following retrograde progression [53,54,55]. In physiological conditions (Figure 1), α-syn monomers allow neurotransmitter release through the modulation of synaptic vesicle maturation and release [46]. However, under pathological conditions, α-syn monomers undergo post-translational modifications, developing toxic α-syn oligomers and fibrils, which are considered the most toxic species of α-syn [56]. These toxic α-syn fibril aggregates determine the onset of synapse deconstruction, and subsequent neurodegeneration [17,57,58]. The first evidence that axons are affected in PD came from the pioneer study performed by Braak and collaborators. The authors were able to demonstrate that α-syn inclusions were not only present at the level of neuronal soma, but also in axonal processes [59]. In addition, in 2003, they demonstrated that extracellular misfolded α-syn could propagate from neurons to glial cells in a prion-like manner [60]. Moreover, Kordower and collaborators [61] discovered the presence of α-syn pathology in dopaminergic neurons implanted about 14 years earlier in PD patients, confirming Braak’s theory. Investigations in animal models and in vitro experiments corroborated the prion-like propagation hypothesis. Pre-formed fibrils (PFFs) of α-syn, in cell culture [62] or injected in rodent brains, engaged endogenous α-syn, producing pathologic inclusions and/or replaying PD features, such as inflammation and DA degeneration [27,63,64,65,66,67].

Specifically, α-syn is localised to presynaptic boutons and is considered as a causative factor for the onset of PD. During the early stage of the aggregation process, it affects synaptic terminals, preceding the loss of striatal dopaminergic neurons, which is considered the most important feature of neurodegeneration in the SN [3,17]. The high association of α-syn with the synaptic vesicles suggests its physiological involvement in the regulation of synaptic transmission, as well as in almost every step of synaptic vesicle recycling, including trafficking, docking, fusion, and recycling after exocytosis [57,68,69]. In fact, alterations in striatal dopamine (DA) release are not correlated with the total amount of DA on striatal tissue or to cell death. Rather, they seem to be caused by functional impairment of neurotransmitter release at the synapse [70]. Garcia-Reitbock and collaborators [71] have demonstrated that in a transgenic α-syn (1–120) mouse model, the protein aggregates were present at striatal dopaminergic terminals and the release of DA from nigrostriatal synaptic terminals was progressively impaired, with nigral dopaminergic neuron loss.

### 2.2. α-Synuclein Induces Protein Synapse Dysfunctions

Accumulation of pathologic α-syn leads to reductions in synaptic proteins, and progressive impairments in neuronal network function and connectivity, causing neuron death. Burrè and collaborators [72] have shown that α-syn cooperates with cysteine-string protein α (CSPα) to promote SNARE complex assembly and support its folding through direct binding to vesicle-associated membrane protein 2 (VAMP-2)/synaptobrevin 2 on synaptic vesicles. In physiological conditions (Figure 1), α-syn is also important for the maintenance and redistribution of the SNARE complex, which is directly involved in neurotransmitter release [54,72]. Accumulated α-syn alters the levels and localization of SNARE proteins at the presynaptic nerve terminals. These findings suggest that α-syn accumulation causes toxicity to synaptic functions, impairing their connectivity. The presence of large α-syn oligomers inhibits SNARE complex formation, blocking vesicle docking, which is an important process for physiological exocytosis [73]. In 2011, Volpicelli-Daley and collaborators demonstrated, in an in vitro α-syn toxicity model, that the incubation of α-syn fibrils on primary neuronal cultures caused loss of the synaptic vesicle proteins SNARE complex, VAMP2, and synaptosomal-associated protein-25 (SNAP-25), as well as CSPα and synapsin-2 [62]. The researchers further investigated the impact of this preformed α-syn fibrils accumulation on neural network activity. They found that impaired hippocampal network activity occurred prior to the reduction of synaptic protein levels, showing the key role of pathological α-syn on neuronal coordination and connectivity [62]. Therefore, the interactions between aggregated α-syn and synaptic proteins could be considered the molecular basis for functional deficits of synapses in α-synucleinopathies.

### 2.3. α-Synuclein Induces Synaptic Dysfunction

In PD, synaptic dysfunction precedes neurodegeneration, and in the last twenty years, research has focused on identifying early synaptic alterations, considering not only the presynaptic aspect but also the postsynaptic. Impairment of DA transmission in the hippocampus is not attributed to the loss of dopaminergic cells, but to functional release changes and to the lack of long-term synaptic plasticity—such as LTP—expression, accompanied by a decrease in synaptic NMDA receptors and the GluN2A/GluN2B receptor subunit ratio, caused by the early accumulation of α-syn in the presynaptic terminals [25].

Several studies have indicated that both GluN2A- and GluN2B-containing receptors can be located in either synaptic or extrasynaptic compartments [74,75,76], and the overexpression of α-syn increases the phosphorylation of these glutamate receptor subunits [77]. Subsequent studies focused on analysing the impact of α-syn inclusion formation on excitatory hippocampal neuronal function at different time points, to verify if its interference precedes neuron death. Froula and collaborators [78] found that the formation of protofibril α-syn inclusions increased the frequency of miniature excitatory presynaptic currents action potential-independent (mEPSCs), with an increased number of docked presynaptic vesicles, despite a reduction in dendritic spine density. On the other hand, the frequency or amplitude of spontaneous excitatory postsynaptic currents (sEPSCs) was unaffected [78]. Therefore, these data suggest there are underlying compensatory mechanisms that preserve normal levels of spontaneous synaptic activity driven by action potentials. In hippocampal slices, α-syn oligomers, but not monomers or fibrils, impair LTP and increase basal synaptic transmission through a mechanism dependent on NMDA receptor activation, triggering an enhanced contribution of GluA1-containing α-amino-3-hydroxy-5-methyl-4-isoxazolepropionic acid (AMPA) receptors [26]. Interesting studies have demonstrated that the overexpression of C-terminally cleaved human α-syn in mice induces several neuropathological PD changes that are not related to dopaminergic neuron loss [25,79]. Navarria and colleagues [80] have shown a functional interaction between α-syn and NMDA receptors, notably with the GluN2B receptor subunit, which is particularly involved in PD [74,81]. The data showed that the membrane translocation and function of GluN2B can be negatively modulated by α-syn, contributing to neuronal degeneration.

Moreover, α-syn is highly expressed in cholinergic neurons of the *pedunculo pontine nucleus* and basal forebrain, in which its alterations in the neural network could lead to cognitive deficits. α-syn is also expressed in GABAergic presynaptic terminals of the globus pallidus and *Substantia Nigra pars reticulata* (SNpr), arising probably from spiny projection neurons (SPNs) of the striatum [82].

The striatum is a subcortical nucleus particularly involved in PD, and it receives strongly excitatory inputs from the cortex and the thalamus [83]. Several studies have demonstrated that the oligomeric forms of α-syn affect synaptic transmission of different neuronal subtypes or synaptic sites, causing motor and behaviour deficits typical of PD [27,28,84]. Tozzi and collaborators [28] have recently shown that the LTP of excitatory synaptic transmission of striatal cholinergic interneurons is compromised by incubation of a nanomolar dose of exogenous human α-syn oligomers (3 nM) on striatal rat slices. The authors linked the loss of LTP in striatal cholinergic interneurons with the close interaction of α-syn and the reduction of GluN2D-mediated NMDA receptor currents. Several years later, the same research group described how the incubation of a high oligomeric α-syn concentration (30 nM) reduced NMDA receptor-mediated synaptic currents and impaired the corticostriatal LTP of striatal SPNs. Furthermore, the treatment of the striatal slices with antibodies targeting α-syn prevented both the α-syn-induced loss of LTP and the reduced synaptic localisation of the GluN2A NMDA receptor subunit [27]. The result of these studies was the awareness that increasing α-syn oligomer concentrations progressively affect NMDA receptor-mediated synaptic plasticity in distinct neuronal populations, indicating that vulnerability to this protein is cell- and region-specific. Another interesting study by Trudel and colleagues [84] recently affirmed that α-syn oligomers induce Ca^2+^-dependent release of glutamate from astrocytes and found that mice overexpressing α-syn manifest increased tonic release of glutamate in vivo. This extracellular glutamate release activates specific receptors, including extrasynaptic NMDA receptors (eNMDARs), and oligomerised α-syn can directly activate these receptors, contributing to neuronal damage.

## 3. α-Synuclein and Its Role in Neuroinflammation in Parkinson’s Disease

As mentioned above, α-syn represents a pathological hallmark of PD, in particular intraneuronal inclusions known as LBs and/or LNs [49]. During brain physiological activity, α-syn is specially found within the presynaptic terminal of neurons belonging to the neocortex, striatum, hippocampus, thalamus, and cerebellum [85,86,87], whose cellular function has not yet been clarified, even if its involvement in synaptic plasticity and in the release of neurotransmitters and synaptic vesicles has been recognised, as extensively discussed in the previous section of this review.

The *SNCA* gene encodes α-syn, and can undergo missense mutations (A53T, A30T and E46K) and multiplication (duplications and triplications), which demonstrates the key role of this protein in PD [6,88,89,90,91,92,93,94]. As previously mentioned, under such pathological conditions, α-syn can be overexpressed and acquire a misfolded conformation. These misfolding species can accumulate, because of impaired autophagy or reduced phagocytic clearance [85,95,96], and could assume aggregated forms, such as oligomers or protofibrils, that cause acute toxicity in the brains of PD patients [97,98]. α-syn aggregates to resist degradation and to prompt, as shown in in vitro experiments, the impairment of macroautophagy, reducing autophasome clearance and promoting dopaminergic neuron death [99]. Moreover, post-translational modifications (ubiquitination, nitration, and phosphorylation) facilitate the formation of these aggregate species, increasing the disease process [85,96,100,101]. However, α-syn is not only a citoplasmatic protein, but can also be found in the extracellular space [102]. Neuronal cells normally throw out α-syn in the extracellular space, which impairs biological processes, such as oxidative stress or mitochondrial and lysosomal dysfunction, amplifying its release [103,104,105,106,107,108]. Thus, extracellular α-syn, probably secreted through exosomes and exocytotic vesicles [103,104,105,106,107,108], could act as damaged-associated molecular patterns (DAMPs), triggering the immune system and neuroinflammation processes [37,109]. As previously indicated, PFF aggregates could activate endogenous α-syn, triggering inflammatory and neurodegenerative responses and, in the final instance, PD pathology [27,63,64,65,66,67]. Thus, in the context of nigrostriatal degeneration in PD, misfolded α-syn species could be considered as being strictly associated with neuroinflammatory events, acting as an antigen capable of activating immune molecules.

## 4. Innate Immunity

The innate immune system represents the first line of defence in the CNS against cellular debris, protein aggregates, and foreign invading pathogens. It acts by non-specific mechanisms and triggers the adaptive system to help fight. Interestingly, immune cells (microglia, monocytes, NK cells, B cells, T cells) express many of the genes involved in the genetic risk of PD, including *SNCA*, *LRRK2*, *DJ*-*1*, and *Parkin* [110].

### 4.1. Microglia

Microglia are the largest population of resident immune cells in the brain. These macrophage-like cells have many functions in the CNS [97,111,112]. First, they monitor the surrounding environment with their elongated processes to protect it against non-self-agents. Moreover, microglial cells also have important functions in brain development, controlling neuron survival, apoptosis of neuronal subpopulations, synaptic pruning, maturation, and regulating the number of synapses. Microglia influence synaptic transmission and plasticity through several mechanisms and “eat me” signals expressed by neurons, such as chemokine CX3CL1 and the classical complement proteins C1q and C3 [83]. Moreover, microglia produce soluble immune agents—Brain-derived neurotrophic factor (BDNF) and IL-1b—capable of modifying synaptic plasticity (Figure 1). Lastly, the production of the oxidative stress mediator dihydronicotinamide-adenine dinucleotide phosphate (NADPH) oxidase induces the alteration of neuronal properties [83].

Microglia express phagocytic activity by deleting dead neurons, protein aggregates, synapses, pathogens, and other antigens that could damage the CNS [97]. These agents are defined as pathogen-associated molecular patterns (PAMPs), or DAMPs [92], which activate microglia. Microglia then change their morphology, moving from the “resting” state, characterised by a round cell body and long and thin processes, to the “activated” state, in which they assume a large cell body and amoeboid shape, with thicker extensions [113,114]. Moreover, this latter phenotype, named M1, exhibits an increased number of major histocompatibility complex (MHC-I) and MHC-II molecules, and amplified levels of pro-inflammatory species, such as chemokines and cytokines (IL1, IL6, TNF). M1 microglial cells affect blood brain barrier (BBB) homeostasis, causing infiltration of peripheral immune cells and increased inflammation [97,111,112]. Interestingly, there is an alternative activated state, termed the M2 microglia, that, on the contrary, expresses anti-inflammatory agents, such as cytokines (IL4 and IL10), pro-resolving mediators, and resolvins [111].

Understanding of α-syn involvement in the activation of microglia originated from a study that provided the first evidence of an inflammatory response in the BV2 cell line [112,115]. Different research groups have demonstrated the release of proinflammatory factors from microglia following injections of PFFs of synthetic α-syn [109,112,116]. The activation of microglia by α-syn increases both microglial phagocytic activity and the spread of α-syn in a prion-like manner [97,117,118,119]. As described previously, extracellular α-syn activates microglia through toll-like receptor 2 (TRL2), acting as a DAMP, and the presence of fibrillar α-syn induces activation of NF-kB [37,109]. This mechanism is very important for the microglia inflammatory response, and also the NLRP3 (nucleotide-binding oligomerization domain leucine-rich repeat and pyrin domain-containing protein 3) inflammasome [112]. NLRP3 induces the production of IL-1 and IL-18 pro-inflammatory cytokines [37] through caspase 1 proteolytic activity [112]. Moreover, caspase 1 activation causes truncation of α-syn, which forms more aggregates, increasing inflammation [112,120]. However, the evidence regarding NLRP3 in PD is still limited, and further detailed studies are required.

### 4.2. Astroglia

Like microglia, astrocytes play a crucial role in neuroinflammation. They represent the most abundant type of cell in the brain and are identified through their recognisable star shape with elongated extensions [111,121,122]. They are essential to maintain homeostasis in the CNS, through the structural support of neuronal cells, preservation of the permeability of the BBB, release of neurotrophic factors, clearance of neurotransmitters, synapse formation and pruning, phagocytic activity, and involvement in synaptic plasticity [111,121]. Indeed, they form the “tripartite synapse”, in which astrocytes participate actively in the exchange of synaptic information between the pre- and post-synapses, regulating plasticity through the release of gliotransmitters [123,124] via a Ca^2+^-dependent release.

Many studies have highlighted the reaction of astrocytes in in vitro and in vivo models of PD, as well as in post-mortem brains of PD patients [85,125,126,127,128,129,130].

In disease conditions, when astrocytes are activated, they change their morphology and gene expression. There are two types of astrocytes: A1 reactive and A2 neuroprotective types. A1 type astrocytes represent reactive microglia-induced cells in stress or disease conditions and are induced by the release of IL-1, TNF and C1q [111,121]. These reactive cells are detected in injured brain regions, where they lose normal functions, release proinflammatory factors (IL-1, IL-1, and TNF), contribute to inflammatory neurodegeneration, and increase their expression of a cytoskeletal protein called Glial Fibrillary Acid Protein (GFAP) [121,131]. However, there are contradictory results from experiments conducted on animals or the post-mortem brains of PD patients. Studies on human PD tissues have displayed null or mild increases of astrocytes and/or GFAP immunoreactivity [131,132,133,134,135]. On the contrary, Parkinsonian animal models undergoing 1-methyl-4-phenyl-1,2,3,6-tetrahydropyridine (MPTP) or 6-hydroxydopamine (6-OHDA) treatment have exhibited dramatic astrogliosis [131,136,137].

A2 astrocytes are defined as a neuroprotective type and secrete neurotrophic factors, such as nerve growth factor (NGF) and BDNF [111,138]. As mentioned above, the A1 astrocyte population is a feature of aging and neurodegenerative diseases, such as PD and Alzheimer’s disease (AD). Phagocytic activity is one of the normal functions lost by A1 reactive astrocytes. This astrocytic activity is important to delete neuronal debris and protein over-expression. Through this process, astrocytes can “ingest” a considerable quantity of α-syn fibrils, as demonstrated by in vitro studies [121,139]. Notably, astrocytes participate in the spreading of α-syn, which can be released from neurons, taken up by astrocytes, and transferred to another astrocyte via extracellular vesicles or exosomes [70,121,140]. In addition, astrocytes have neurotransmitter receptors and Na^+^-dependent excitatory amino acid transporters, such as glutamate transporter 1 (GLT-1), excitatory amino acid transporter 2 (EAAT2), and glutamate aspartate transporter (GLAST). Furthermore, they participate in glutamate re-uptake and release, defending neurons from glutamate spill over and excitotoxicity [131,141]. Moreover, proinflammatory factors, secreted by microglia, affect glutamatergic signals, e.g., reducing the expression of glutamate transporters [131,142,143].

## 5. Adaptive Immunity

As seen previously, neuroinflammation is a complex cerebral phenomenon that involves not only innate immunity, but also the adaptive immune system, arising from periphery due to the alteration of the BBB [141,144,145]. In the past, the brain was considered a privileged organ which was not susceptible to infiltration by peripheral leukocytes, through BBB activity [110,115,146]. However, in the last few years, many research groups have highlighted the presence of T cells in the post-mortem brains of people with neurodegenerative disorders, such as in PD patients or in experimental animal models [64,110,147].

Positron emission tomography (PET) diagnostic analysis in PD patients has pointed out the close relationship between BBB alteration, infiltration of CD^+^4 and CD^+^8 lymphocytes, and neuronal loss [111,148].

Indeed, microglia seem to induce the activation of T cells, acting as antigen presenting cells (APCs). Microglia activate CD4+ expressing antigen through MHCII. MHCI activates CD8+ lymphocytes, and is expressed by all cells of the organism [149]. CD4+ species, recruited from the periphery, infiltrate the CNS and can exhibit both an anti- (Th2 and Tregs) or pro-inflammatory phenotypes (Th1 and Th17). During PD progression with dopaminergic cell loss, microglia and immune cells (CD+4 cells, macrophages) undergo alteration of the neurotransmitter balance (DA levels are reduced in SNpc and cortical glutamatergic transmission is over-expressed), increasing neuroinflammation [150].

## 6. α-Synuclein and Glia Cells in Parkinson’s Disease Synaptopathy

Neuronal transmission at corticostriatal synapses is altered in PD [151], due to the impaired afflux of DA in striatum that leads to LTP and LTD loss [151]. In this scenario, synaptopathy seems to strictly be linked to α-syn aggregated species [83]. Moreover, the immune system seems to be involved in the α-syn-induced PD synaptopathy. Resident immune cells in the brain may regulate synaptic transmission. As seen above, microglia and astrocytes monitor the surrounding environment, but when activated, for example by α-syn aggregation, they produce cytokines, chemokines, neurotransmitters, and growth factors, through which they can modulate neuronal plasticity. Indeed, many studies have demonstrated an increase of cytokine levels in the striatum and CSF of Parkinsonian patients. Interestingly, these molecules are involved in the synaptic plasticity process [151].

Moreover, most of the immune species, belonged to both innate and adaptive systems, expressed dopaminergic receptors and the necessary machinery to synthesize, metabolize and store up DA [76,111,152]. Under physiological conditions, in the nigrostriatal pathway, in which the DA concentration is normally high, this could stimulate low-affinity receptors (D1R and D2R) and exert anti-inflammatory effects [76]. When DA levels are decreased, as shown in PD, there is a stimulation of high-affinity DA receptors (in particular, DA D3R), which can have pro-inflammatory effects that lead to inflammation and consequent neurodegeneration in the brain [76]. Remarkably, Duffy and collaborators, in 2018, studied the activation of microglia after synthetic α-syn-PFF unilateral injections into the striatum. They found out that microglia reactions occurred months before cell death in the SNpc [153], suggesting early activation of the immune system in the brain during PD progression. Additionally, astrocytes are activated by α-syn misfolding, inducing chemokine and cytokine production via TRL4. Moreover, α-syn leads to neurotoxicity following Ca^2+^ flux and oxidative stress [96].

All together, these findings represent a complex scenario in which α-syn aggregation triggers the neuroinflammatory process, which in turn modulates synaptic activity through the production of soluble immune molecules.

## 7. Neuroinflammation as a Therapeutic Target in Parkinson’s Disease

Neuroinflammation is considered an early event in the pathophysiology of PD, or a driving mechanism of its progression. Thus, it could be an early therapeutic target to counteract PD. Many recent studies have described the use of different agents that act on inflammatory species [97,111]. In the past few years, it has been hypothesized that there is a correlation between the chronic use of non-steroidal anti-inflammatory drugs (NSAIDs) and reduced risk of developing PD [29,154]. Different meta-analyses have presented conflicting data. In particular, some studies did not find any correlation between the use of anti-inflammatory drugs and PD risk [155,156]. Other research studies showed a beneficial outcome for the use of non-aspirin NSAIDs [157,158], or aspirin in females [159]. Growing evidence supports the importance of neuroinflammation as a therapeutic target, with the aim of limiting disease progression.

Several clinical trials have been conducted in the last few years in this field (https://clinicaltrials.gov (accessed on 10 May 2021)). Immunotherapies against α-syn have been developed to reduce extracellular α-syn levels, which are responsible for triggering neuroinflammation, and its spread. There are two types of immunotherapies: active and passive [160,161].

Active immunotherapy, or vaccination, involves the production of antibodies against α-syn through the patient’s immune system. AFFITOPE^®^ AFF1 belongs to this category, and is characterised by administration of small fragments of α-syn. This immunotherapy has successfully reduced neurodegeneration and increased anti-inflammatory factors in two animal models [111]. AFFITOPE^®^ AFF1 is currently being considered in a phase I clinical study (NCT02267434) [111,162].

Passive immunisation involves the administration of antibodies against α-syn, and has achieved good results in pre-clinical studies by increasing the clearance of α-syn [162]. A phase 2 clinical trial that uses PRX002 as a passive immunotherapy [111,162] is currently underway.

Immunotherapies could be a valid target for therapeutic approaches, either used in combination with current medications or even as an alternative treatment in the future.

## 8. Conclusions

Synaptic damage and neuronal loss are major neuropathological features of PD. Misfolded α-syn aggregates are associated with disease progression; this is known as synucleinopathy. In the non-pathological brain, α-syn is not toxic and participates in several functions associated with neurotransmission and synaptic plasticity, including synaptic vesicle recycling and neurotransmitter synthesis and release. Alterations to the conformity of α-syn lead to the beginning of pathological processes in synucleinopathies. To date, many studies have identified the functions of α-syn in the regulation of neurotransmission and synaptic plasticity, but further insights are needed to outline its pathological roles. Synaptopathy, α-syn misfolding/aggregation, and neuroinflammation processes seem to interact and contribute to PD pathogenesis. As demonstrated by many of the studies cited above, these mechanisms affect each other, creating a vicious circle in which it is difficult to establish the first pathology trigger. The involvement of α-syn pathology and an altered immune response point to potential new immunomodulatory targets to slow down the progression of the disease, and/or improve its outcome.

## Figures and Tables

**Figure 1 ijms-22-06517-f001:**
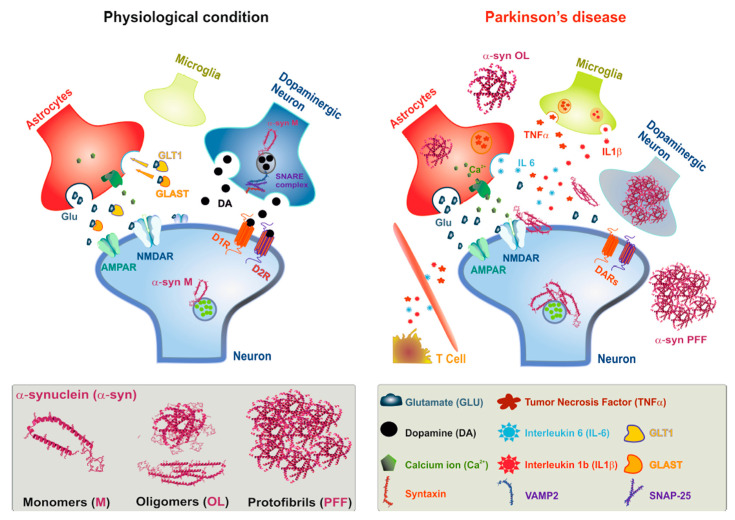
The presynaptic exo-endocytotic cycle is specifically affected in α-syn-related synaptopaties. α-syn accumulation in the presynaptic terminals causes synaptopathy, ultimately leading to neurodegeneration. Under physiological conditions, α-syn, as a monomer, acts at the presynaptic terminals and activates a molecular machinery relevant to synaptic transmission, including the SNARE complex proteins, which recruit synaptic vesicles. After docking and priming, the vesicles undergo SNARE-mediated membrane fusion at the active zone, leading to neurotransmitter release into the synaptic cleft. Formation of toxic α-syn species in Parkinson’s disease (PD), such as oligomers and fibrils, has been shown to play a pivotal role in PD pathogenesis. These toxic species, which accumulate at the presynaptic terminal, lead to altered levels of proteins involved in synaptic transmission, determining synaptic dysfunction. Toxic α-syn accumulation affects the synapses, causing a lack of long-term synaptic plasticity expression, followed by an increase in the phosphorylation of glutamate receptors and a decrease in the GluN2A/GluN2B receptor subunit ratio, contributing to neuronal degeneration. Toxic aggregated α-syn forms participate in inflammatory processes mediated by microglia and astrocytes, which lose their normal physiological functions. Microglia become activated (M1) and trigger an immune response through increased receptor expression (MHCI), secretion of pro-inflammatory cytokines and chemokines, and inducing an astrocyte reaction (A1 type). Reactive astrocytes decrease glutamate uptake and release pro-inflammatory mediators. Finally, peripheral immune cells (T cells and monocytes) raid brain tissue and amplify neuroinflammation, contributing to neurodegeneration.

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
