# Peer review of "Alpha-Synuclein as a Prominent Actor in the Inflammatory Synaptopathy of Parkinson’s Disease"

_ijms, 2021, doi:10.3390/ijms22126517_

Round 1
Reviewer 1 Report
This is a succinct review on an important issue of early changes occurring in the synapses in Parkinson's disease. The review is a timely update on this important issue.
I only have minor comments about re-checking the English expression in places and some formatting issues (ie spaces between words).
Author Response
We are grateful to the Reviewer for this comment and for her/his valuable suggestion. We have re-checked the English expression and the text formatting. Moreover, the MDPI Revision English service edited all the text.
Reviewer 2 Report
The manuscript by Cardinale et al. attempts to review the role of alpha-synuclein as an inflammatory agent and its effect on synapses. The review article is scarcely systematic as the pro-inflammatory properties of alpha-synuclein have not been reviewed adequately. The manuscript is hard to read. The authors are encouraged to reorganize the manuscript for a cognizant read.
Major comments
There is inadequate impetus given to elaborate on alpha synuclein-mediated neuronal inflammation. As a result, the section of synaptopathy feels stretched.
The manuscript is not a good read. There are too many grammatical errors and typos to incorporate in this report. Therefore, the authors have to revise the manuscript thoroughly.
References are not cited appropriately. There are several instances where statements are not supported by references Eg. Line 136, line 276.
Minor comments
The physiological localization of alpha-synuclein is not exclusive to presynaptic terminals as it has also been found in mitochondria-associated membranes and the nucleus.
Space is missing in between words throughout the manuscript. E.g., Page91 "Snpcand". There are several instances like this throughout the manuscript.
Expansions should occur in the first instance. Eg. Dopaminergic neurons -DA line90
Several unconventional usages hinder the flow of reading. E.g., "phagocytosing" Line60
Author Response
The manuscript by Cardinale et al. attempts to review the role of alpha-synuclein as an inflammatory agent and its effect on synapses. The review article is scarcely systematic as the pro-inflammatory properties of alpha-synuclein have not been reviewed adequately. The manuscript is hard to read. The authors are encouraged to reorganize the manuscript for a cognizant read.
R: We thank the Reviewer for these critical and constructive comments.
Major comments
There is inadequate impetus given to elaborate on alpha synuclein-mediated neuronal inflammation. As a result, the section of synaptopathy feels stretched.
R: We understand the points of the Reviewer and did our best to ameliorate the description of the alpha synuclein-mediated neuronal inflammation. We have now discussed this argument at the 6th paragraph at pages 15-16.
The manuscript is not a good read. There are too many grammatical errors and typos to incorporate in this report. Therefore, the authors have to revise the manuscript thoroughly.
R: We have re-checked the english expression and the text formatting. Moreover the MDPI Revision English service edited all the text.
References are not cited appropriately. There are several instances where statements are not supported by references Eg. Line 136, line 276.
R: The text has been revised accordingly. Please, note that Line 136 refers to the legend of Figure 1.
Minor comments
The physiological localization of alpha-synuclein is not exclusive to presynaptic terminals as it has also been found in mitochondria-associated membranes and the nucleus.
R: We better denoted this concept referring the alpha-synuclein localization. The sentence has been changed as follows: “Synaptopathy is linked to α-synuclein (α-syn), a small soluble protein, encoded by the SNCA gene on human chromosome 45, 6, physiologically mainly localized in the presynaptic nerve terminals7, mitochondrial-associated membrane (MAM)8, in which its overexpression increased the extent of contact sites and downregulated MAM activity9-12, and nucleus13-16” (please, see page 4).
Space is missing in between words throughout the manuscript. E.g., Page91 "Snpcand". There are several instances like this throughout the manuscript.
R: We have re-checked the text formatting.
Several unconventional usages hinder the flow of reading. E.g., "phagocytosing" Line60
R: According to your suggestion, we changed the term "phagocytosing" with “phagocytic”.